# Exploring Microorganisms from Plastic-Polluted Sites: Unveiling Plastic Degradation and PHA Production Potential

**DOI:** 10.3390/microorganisms11122914

**Published:** 2023-12-03

**Authors:** Diana A. Garza Herrera, Marija Mojicevic, Brana Pantelic, Akanksha Joshi, Catherine Collins, Maria Batista, Cristiana Torres, Filomena Freitas, Patrick Murray, Jasmina Nikodinovic-Runic, Margaret Brennan Fournet

**Affiliations:** 1PRISM Research Institute, Technological University of the Shannon Midlands Midwest, N37HD68 Athlone, Ireland; a00278551@student.tus.ie (D.A.G.H.); margaret.brennanfournet@tus.ie (M.B.F.); 2Institute of Molecular Genetics and Genetic Engineering, University of Belgrade, Vojvode Stepe 444a, 11042 Belgrade, Serbia; branapantelic@imgge.bg.ac.rs (B.P.); jasmina.nikodinovic@imgge.bg.ac.rs (J.N.-R.); 3Shannon Applied Biotechnology Centre, Midwest Campus, Technological University of the Shannon, V94EC5T Limerick, Ireland; akanksha.joshi@tus.ie (A.J.); catherine.collins@tus.ie (C.C.); patrick.murray@lit.ie (P.M.); 4UCIBIO—Applied Molecular Biosciences Unit, Department of Chemistry, School of Science and Technology, NOVA University Lisbon, 2829-516 Lisbon, Portugal; mr.batista@campus.fct.unl.pt (M.B.); c.torres@fct.unl.pt (C.T.); a4406@fct.unl.pt (F.F.); 5Associate Laboratory i4HB—Institute for Health and Bioeconomy, School of Science and Technology, NOVA University Lisbon, 2829-516 Lisbon, Portugal

**Keywords:** screening, plastic degradation, upcycling, biopolymers, polyhydroxyalkanoates

## Abstract

The exposure of microorganisms to conventional plastics is a relatively recent occurrence, affording limited time for evolutionary adaptation. As part of the EU-funded project BioICEP, this study delves into the plastic degradation potential of microorganisms isolated from sites with prolonged plastic pollution, such as plastic-polluted forests, biopolymer-contaminated soil, oil-contaminated soil, municipal landfill, but also a distinctive soil sample with plastic pieces buried three decades ago. Additionally, samples from *Arthropoda* species were investigated. In total, 150 strains were isolated and screened for the ability to use plastic-related substrates (Impranil dispersions, polyethylene terephthalate, terephthalic acid, and bis(2-hydroxyethyl) terephthalate). Twenty isolates selected based on their ability to grow on various substrates were identified as *Streptomyces*, *Bacillus*, *Enterococcus*, and *Pseudomonas* spp. Morphological features were recorded, and the 16S rRNA sequence was employed to construct a phylogenetic tree. Subsequent assessments unveiled that 5 out of the 20 strains displayed the capability to produce polyhydroxyalkanoates, utilizing pre-treated post-consumer PET samples. With *Priestia* sp. DG69 and *Neobacillus* sp. DG40 emerging as the most successful producers (4.14% and 3.34% of PHA, respectively), these strains are poised for further utilization in upcycling purposes, laying the foundation for the development of sustainable strategies for plastic waste management.

## 1. Introduction

Over the past few decades, there has been significant interest in identifying microorganisms capable of degrading plastic and the enzymes involved in plastic degradation. To date, only a limited number of microbial agents and their enzymes have been recognized and studied for plastic degradation, yet with minimal effectiveness. This could be attributed to the prevalence of uncultivated microbial species, which consequently remain unexplored in the context of the respective plastic-degrading environment [1]. Despite the high-molecular-weight polymers (e.g., polypropylene (PP), polystyrene (PS)), potential degraders are widely spread across the microbial tree of life, and evidence for plastic degradation by a majority of these taxa is still limited [2]. However, when it comes to polyethylene terephthalate (PET), various bacteria (e.g., *Thermobifida fusca*, *Thermomonospora curvata*, and *Ideonella sakaiensis*) [3,4,5] and fungi (e.g., *Fusarium solani, Humicola insolens,* and *Aspergillus oryzae*) [6,7,8] have been proven to possess enzymatic machinery to degrade this polymer. 

Microorganisms have been exposed to conventional plastics only recently, allowing limited time for their evolution. However, microbes capable of breaking down plastic can be found in various environments, such as soil, seawater, and plastic-polluted sites [9]. Landfill sites exhibit significant heterogeneity owing to the complexity of substrates and the abundance of organic and inorganic matter, making them a recognized reservoir of microbial diversity [10]. Hence, many research groups are reporting results on plastic degraders discovered in environments involving severe or moderate plastic pollution [11,12,13]. Ever since Bowditch et al. [14]. reported the ability of Lepidoptera and Coleoptera to partially degrade polyvinyl chloride and PP packaging films, the scientific community has turned their attention to this unconventional source of plastic degraders. Kim et al. [15] found bacteria in the intestines of adult bees with the ability to degrade PET, while Pham et al. [16] reported the ability of mealworm microbiota to degrade various consumer plastics. The existing information regarding the involvement of insects and their microbiota in plastic decomposition is still quite restricted, leaving numerous questions unanswered about the intricate process of plastic breakdown facilitated by insects. The specific mechanisms and the role of enzymes in this degradation process are yet to be conclusively determined. However, insects certainly do exhibit the capability to degrade compounds that are typically challenging and potentially can find practical applications in waste management programs.

In recent years, notable attention and interest in bioplastics have risen, reflecting a worldwide trend toward the preference for biodegradable products by consumers. Bioplastics stand out due to their inherent biodegradability, sustainability, and environmentally friendly properties, presenting numerous advantages over conventional petrochemical-based plastics [17]. However, it is important to point out that not all biodegradable materials are biobased. Biobased materials are derived from renewable resources, from living (or once-living) organisms, while biodegradable materials possess the ability to decompose in a given environment over a reasonable amount of time [18]. The most extensively studied materials, both biobased and biodegradable, are polyhydroxyalkanoates (PHAs). These materials are poised to replace certain contemporary petrochemically derived plastics due to their biodegradable nature and favourable thermoplastic and mechanical properties. These include versatility, elasticity, flexibility, and other desirable characteristics [19]. PHA production using waste streams from several industries, municipal waste, and households is already well examined and established [20,21]. 

Nevertheless, recent studies have indicated that plastic waste can be used for the same goal [22,23]. The biological process of plastic upcycling involves utilizing plastic waste streams as a carbon substrate for biotechnological processes, following a methodology employed for lignocellulosic feedstocks [24]. Numerous enzymes with hydrolytic properties have been identified, capable of depolymerizing specific plastics with the possibility or specifically engineered to transform depolymerized plastics into high-value products, such as biopolymers [22,25,26]. However, the effectiveness of the biological route is often impeded by the inherent recalcitrance of plastics, typically associated with hydrophobicity and crystallinity. 

Presently, recycling technologies predominantly yield lower-value products from plastic waste (downcycling) or, at best, products of similar value. To propel the development of technologies facilitating plastic upcycling, the European Commission has allocated significant funding to research and innovation projects, encompassing both chemical and biological processes. Bio innovation of a circular economy for plastics (BioICEP) is one of EU-funded projects tackling plastic waste management and upcycling. The overarching aim of the project is to illustrate a streamlined and sustainable approach toward achieving a circular economy for plastics. This involves the creation of an advanced, energy-efficient, and cost-effective method for biotransforming waste plastic into bioproducts and bioplastics that are in high demand in the market. As a part of the BioICEP project, the first steps of this study involved the creation of microbial collection from soil samples tainted with plastic waste and samples gathered from two types of insects. Based on the literature reported earlier in this section, this approach was selected to increase the probability of isolating microbes equipped with the requisite enzymatic machinery for plastic degradation. Overall, this study delved into the microbial diversity of plastic-polluted sites, exploring their proficiency in degrading various plastic-related substrates as sole carbon and energy sources. The ability of these strains to produce high-value molecules such as biosurfactants and PHA was analysed. Furthermore, the confirmed capacity of selected strains to utilize plastic-related substrates and generate biopolymers added depth to our understanding of their ecological roles in plastic degradation and possible use for plastic upcycling.

## 2. Materials and Methods

### 2.1. Collection of Samples

Various sampling sites were chosen to be examined as a source of microorganisms with potential ability to degrade plastic-related substrates, such as plastic-polluted sites, landfills, oil-polluted sites, but also, arthropods (isopods and arachnids) were collected as a sampling material. The number of sites, their origin, and nature are presented in Table 1. 

### 2.2. Bacterial Isolation

Microbial isolation was carried out via the serial dilution method using different selective media in order to facilitate various microbial species growth. Thus, 1 g/L of samples was used as a starting point. For arachnids and isopods, samples were washed with 70% (*v*/*v*) ethanol, rinsed in distilled water, and followed up with an overnight freeze at −80 °C. Samples were thoroughly macerated, and 1 g/L was used to prepare serial dilutions. For these purposes, Lysogeny broth (LA: 10 g/L tryptone, 5 g/L yeast extract, 10 g/L NaCl, 15 g/L agar), Sabouraud Dextrose (SAB: 40 g/L glucose, 10 g/L peptone, 15 g/L agar), and Tryptic Soy Agar (TSA: 17 g/L casein peptone (pancreatic), 2.5 g/L, dipotassium hydrogen phosphate, 2.5 g/L glucose, 5 g/L sodium chloride, 3 g/L soya peptone, 15 g/L agar) were used. After 10 days of incubation at 30 °C and 37 °C, morphologically distinct strains were selected and separated. Obtained pure cultures were stored in glycerol (20%, *v*/*v*), maintained at −80 °C, and used for the inoculation of cultures for further experiments. Morphological characteristics of selected pure cultures were recorded on TSA plates using Stereo Microscope Olympus szx10, 10× magnification (Boston, MA, USA) [27,28]. 

### 2.3. Agar-Based Screening Methodology 

The ability to utilize different plastic monomers and oligomers as a sole carbon and energy source was investigated using Mineral Salt Medium agar plates (MSM): 15 g/L agar, 9 g/L Na_2_HPO_4_ × 12H_2_O, 1.5 g/L KH_2_PO_4_, 1 g/L NH_4_Cl, 0.2 g/L MgSO_4_ × 7H_2_O, 0.2 g/L CaCl_2_ × 2H_2_O, 0.1% trace elements solution, 0.025% N-Z amine and carbon source 5 g/L. Plastic-related substrates were used as a sole carbon source: terephthalic acid powder (TPA; Sigma Aldrich, London, UK), bis(2-hydroxyethyl) terephthalate powder (BHET; Sigma Aldrich, UK), polyethylene terephthalate powder (PET; Sigma Aldrich, UK), anionic aliphatic polyester-polyurethane dispersion (Impranil DLN-SD, Covestro Ltd., London, UK), and anionic polycarbonate polyurethane dispersion (Impranil DL2077; Covestro Ltd., London, UK), whereas glucose was used as a positive control. The applied method is based on the combination of several references and previous experience [29,30,31].

### 2.4. 16S rRNA Sequencing

Strains selected as the best performers in screening assays were identified via 16S rRNA sequence analysis. These strains were cultivated in tryptone soy broth (TSB) (17 g/L casein peptone (pancreatic), 2.5 g/L, dipotassium hydrogen phosphate, 2.5 g/L glucose, and 5 g/L sodium chloride, 3 g/L soya peptone) at 30 °C and 37 °C for 4 days shaking at 180 rpm for DNA isolation [32]. Strains were identified by 16S rRNA sequence analysis using universal bacterial primer set: 27F (5′-AGAGTTTGATCCTGGCTCAG-3′) and 1492R (5′-GGTTACCTTGTTACGACTT-3′). PCR amplification was performed in a 2720 Thermal Cycler (Applied Biosystems, Thermo Fisher Scientific, Waltham, MA, USA) using Phusion High-Fidelity PCR Master Mix (F-531S) (Thermo Scientific, Waltham, MA, USA) following the manufacturer’s protocol. PCR products were purified using a PCR purification kit (Qiagen, Hilden, Germany). Sequencing was performed using the Sanger sequencing method at Source Bioscience (Cambridge, UK). 16S rRNA sequences were identified using BLASTN program [33].

### 2.5. Phylogenetic Analysis

The taxonomic verification of the isolated bacterial strains was carried out through an examination of the 16S rRNA sequence data. These sequences were compared with those deposited in the GenBank database of NCBI using the BLAST similarity search tool “https://blast.ncbi.nlm.nih.gov/Blast.cgi” (accessed on 10 September 2023). The sequence with the highest total score, substantial percentage query coverage, high identity percentage, and lowest E-value was chosen from all the identified homologous nucleotide sequences. Sequences exhibiting a sequence similarity of >98% to their closest phylogenetic neighbor over an average range of 1300–1400 bp were classified at the species level. Sequences with <98.0% sequence similarity were categorized at the genus level, and the percentage sequence similarity is indicated in brackets. The 16S rRNA sequences were submitted to GenBank and assigned the following accession numbers: OR693299, OR693300, OR693301, OR693302, OR693303, OR693304, OR693305, OR693306, OR693307, OR693308, OR693309, OR693310, OR693311, OR693312, OR693313, OR693314, OR693315, OR693316, OR693317, and OR693318. The evolutionary relationship was inferred using the neighbor-joining method [34]. The optimal tree is shown. The percentage of replicate trees in which the associated taxa clustered together in the bootstrap test (1000 replicates) is shown next to the branches [35]. The tree is drawn to scale, with branch lengths in the same units as those of the evolutionary distances used to infer the phylogenetic tree. The evolutionary distances were computed using the Tamura–Nei method and are in the units of the number of base substitutions per site [36]. This analysis involved 24 nucleotide sequences. Codon positions included were 1st + 2nd + 3rd + Noncoding. All ambiguous positions were removed for each sequence pair (pairwise deletion option). There were a total of 1405 positions in the final dataset. Evolutionary analyses were conducted in MEGA11 [37].

### 2.6. Upcycling Capabilities of Selected Isolates

#### 2.6.1. Emulsification Assay

For emulsification activity determination, 2 mL of cell-free supernatant and 2 mL of sunflower oil were mixed in test tubes. The mixtures were vigorously vortexed for 1 min and allowed to stand for 24 h at room temperature. By adopting the formula given below, the emulsification index (EI, %) was calculated (Equation (1)). Polysorbate 80 was used as a positive control (EI 100%).
(1)$$EI\left[\%\right]=\frac{Height\;of\;the\;emulsified\;layer}{Total\;height\;of\;liquid\;mixture}\times 100$$

#### 2.6.2. Reactive Extrusion

PET depolymerization experiment via reactive extrusion (REX) was conducted on a benchtop PrismTM twin-screw extruder (Thermo Electron GmbH, Karlsruhe, Germany) using a modified procedure to that of Biermann et al. (2021) [38]. Post-consumer PET material (PET tray, provided by Novelplast, Gibstown, Ireland) was mixed with solid NaOH in a ratio (2:1) in a sealed plastic bag, and the mixture was dispensed through the main shaft into the barrel. The temperature of the barrel was kept constant at 250 °C, and the rotational speed of the screws was adjusted to 20 rpm. Unreacted porous polymer residue was used for PHA production experiments as a sole carbon source.

#### 2.6.3. Bacterial Cultivation for PHA Production

The cultures were reactivated from cryopreserved stock cultures (in 20% (*v*/*v*) glycerol, at −80 °C) by plating into solid Luria Bertani (LB) medium (tryptone, 10 g/L; yeast extract, 5 g/L; NaCl, 10 g/L; agar, 15 g/L; pH 7.0). The plates were incubated at 30 °C until isolated colonies formed. The inoculum for the shake flask experiments was prepared by inoculating a single colony in 50 mL liquid LB medium (tryptone, 10 g/L; yeast extract, 5 g/L; NaCl, 10 g/L; pH 7.0) and incubation in an orbital shaker (200 rpm), at 30 °C, for 24 h. For the shake flask assays, modified Medium E* (pH 7.0) was used, with the following composition: (NH_4_)_2_HPO_4_, 1.1 g/L; K_2_HPO_4_, 5.8 g/L; KH_2_PO_4_, 3.7 g/L; 10 mL/L of a 100 mM MgSO_4_ solution and 1 mL/L of a micronutrients’ solution. The micronutrient solution was composed of (per litre of 1 N HCl): FeSO_4_⋅7H_2_O, 2.78 g; MnCl_2_⋅4H_2_O, 1.98 g; CoSO_4_⋅7H_2_O, 2.81 g; CaCl_2_⋅2H_2_O, 1.67 g; CuCl_2_⋅2H_2_O, 0.17 g; and ZnSO_4_⋅7H_2_O, 0.29 g. The medium was supplemented with REX residuals (REX PET) at a concentration of 20 g/L. The assays were performed in 500 mL baffled shake flasks containing 200 mL of Medium E* supplemented with REX PET, which were inoculated with 20 mL of the inoculum prepared as described above. Incubation was performed in an orbital shaker (200 rpm) at 30 °C.

#### 2.6.4. Quantitative and Qualitative Analysis of Derived PHAs 

Cell growth was monitored during the cultivation by measuring the culture broth’s optical density (OD) at 660 nm. The presence of intracellular granules of PHA was assessed by visualization using fluorescence microscopy (BX51, Olympus, Boston, MA, USA) of the cells stained with Nile Blue A (Sigma Aldrich, UK), as described by Silvestre et al. [39]. At the end of the run, the cell pellet was collected by centrifuging the culture broth (10,956× *g*, 15 min, 4 °C) and lyophilized. The resulting dry biomass was used for the gravimetric determination of the cell dry weight (CDW) and for PHA analysis by gas chromatography (GC), as described by Rebocho et al. [40]. PHA content in the biomass and its composition were determined by GC after acidic methanolysis of dried cells. Samples were mixed with 2 mL of a sulphuric acid 20% (*v*/*v*) (Sigma- Aldrich, HPLC grade, UK) solution in methanol (Fisher Chemical, HPLC grade, UK) and 2 mL of benzoic acid (1 g/L) in chloroform (Sigma Aldrich, HPLC grade, UK) solution, at 100 °C, for 4 h. Benzoic acid acted as internal standard. The calibration curve was prepared with standards of P(HB-co-HV) (Sigma Aldrich) composed of 86 mol% 3-hydroxybutyrate (3HB) and 14 mol% 3-hydroxyvalerate (3 HV), at concentrations ranging from 0.1 to 1.0 g/L. The obtained methyl esters were analysed in a gas chromatograph equipped with a flame ionization detector (FID) (Sigma Aldrich, UK) and a Restek column (Crossbond, Stabilwax, Tokyo, Japan) at constant pressure (96 kPa) using helium as carrier gas. Splitless injection was used. The oven temperature ramp was the following: 20 °C/min until 100 °C; 3 °C/min until 155 °C; and, finally, 20 °C/min until 220 °C. The yields of purity obtained for the biopolymers were calculated from Equation (2): (2)$$Polymer\;purity\;\left[\%\right]=\frac{Polymer\;profuced\;\left[g\right]}{Polymer\;analized[g]}\times 100$$

The polymer was extracted from the dried biomass by Soxhlet extraction with chloroform (~5 g biomass for 250 mL chloroform), at 80 °C, for 48 h. Cell debris were removed by filtration with filters with a pore size of 0.45 μm (GxF, GHPmembrane, PALL, UK), and the polymer was precipitated in ice-cold ethanol (1:10, *v*/*v*) under vigorous stirring. The polymer was recovered and dried at room temperature.

#### 2.6.5. Estimation of TPA Content in Cultivation Media by HPLC Analysis

Cultivation broth was centrifuged (13,000 rpm, 15 min) and supernatants were filtered (0.2 μm, VWR) and diluted with 30 mM NaOH in order to assess the amount of TPA. HPLC method was applied using anion exchange column (Ionpac AS11-HC 4.6 × 250 mm equipped with a pre-column) coupled to a conductivity detector. Analysis was performed at 30 °C, and 30 mM NaOH was used as an eluent at the 1.5 mL/min flow rate. TPA standards in concentration range from 0.006 to 1.0 g/L were used for calibration curve construction.

## 3. Results and Discussion

### 3.1. Isolation and Screening of Microbial Strains from Plastic-Polluted Sites

Microorganisms possess inherent or acquired capacities to endure in environments with extreme conditions, such as adaptive changes in transcriptomic, metabolic, and signalling pathways [41]. Adapting to plastic-contaminated surroundings includes biofilm formation [42], utilization of different pathways [43,44], and, predominantly, the secretion of specific depolymerizing enzymes that enable them to use persistent plastic pollution as an additional carbon source, giving them a selective advantage [45]. In this study, we tried to harness the potential of microbial communities to degrade plastic-related substrates by targeting sites where plastic pollution is present for a significant amount of time. A total of seven samples were collected and analysed (Table 1). Samples 1, 5, and 6 were obtained from a plastic-polluted forest, biopolymer-polluted soil, and municipal landfill, respectively. Sample 2 was provided by an Irish farmer who, predicting the plastic pollution problem, buried plastic pieces in soil 3 decades ago. Following up on the concept of insects as a biodegrading tool for synthetic polymers [46], samples 3 and 4 were collected from Arthropoda species: *Oniscidea* (woodlice) and *Hogna aspersa* (garden wolf spiders), respectively. On the other hand, sample 7 was retrieved from oil-polluted soil in Mexico. According to the research performed by Gambarini et al. [2], the majority of presumed plastic degraders were isolated from soil samples (27.8%), followed by plastic waste disposal sites (9.6%) and composts (5.3%). Additionally, a significant portion was sourced from culture collections of microorganisms (15.9%). As illustrated in Figure 1A, the highest number of morphologically diverse isolates, proficient in degrading plastic, was identified in sample 6 from landfill (25%).

This landfill, situated in the small city of Priboj in Western Serbia, poses a significant ecological challenge for the region due to the volume of waste and inadequate disposal and incineration methods. Having been covered with plastic waste for several decades, this area emerged as an ideal starting point in the quest for plastic-degrading microbes. Kopecka et al. (2022) examined isolates from plastic waste deposited in the municipal landfill over 2–17 years and discovered several bacterial strains with the ability to degrade high-density polyethylene (HDPE) [47]. In this study, out of 38 isolates, 23 showed the ability to grow on anionic aliphatic polyester-polyurethane dispersion (IMP DLN-SD) while 16 grew on anionic polycarbonate polyurethane dispersion (IMP DL) as a sole carbon source (Figure 1B). This potential discovery holds promise for addressing polyurethane (PU)-related plastic waste, which constitutes 9% of global plastic production, according to Geyer et al. [48]. A noteworthy proportion of isolates from this source demonstrated the capability to utilize PET and PET-related substrates (TPA and BHET). Another sample that showed interesting potential in this study came from a site constantly polluted by oil, located in Poza Rica, east-central Mexico (Sample 7). Further, 24 strains were isolated in total, out of which 58% showed the ability to grow on impranil and PET. It is interesting to note that none of these strains were capable of using PCL as a carbon source. However, in sample 5 (soil polluted with biopolymers), growth on PCL was detected for five strains. Samples related to insects (samples 3 and 4) did not yield exceptional results, except for five strains displaying the ability to grow on PCL. Sample 1 was derived from a forest mildly polluted with plastics and, thus, considered relatively pristine. Despite isolating 24 different strains, only a handful exhibited potential in utilizing plastic-related substrates, with five strains for BHET, three for PCL, and two for PET.

### 3.2. Most Promising Strains Selection and Characterization

Based on agar screening results, 20 isolates were selected for further investigation, taking into consideration the following factors: their ability to grow on at least three out of five examined substrates as a single carbon and energy source; prominent growth on at least one substrate; incubation period (Table 2). These isolates were further identified through 16S rRNA sequencing, and their morphological characteristics were recorded and summarized (Figure 2).


**
*Sample 1: Plastic-polluted forest*
**


Strain 19 expressed moderate growth on all examined substrates apart from TPA. This strain 16S rRNA sequence (1375 bp length) showed 100% similarity with *Streptomyces* strains according to NCBI Blast Nucleotide base analysis. The *Streptomyces* genus exhibited significant potential in several industries; their enzymatic toolbox and the ability to degrade polysaccharides [49] position them as a compelling candidate for bioremediation and biological control across diverse ecosystems. Results reported by Rodríguez-Fonseca et al. demonstrated the significant potential of *Streptomyces* spp. in the degradation of synthetic polymers, especially polyolephins [50]. *Streptomyces* sp. DG19 exhibits typical behavior of this genus when incubated on the TPA plate, forming compact white colonies with smooth edges [51]. 

Strain 40 expressed the ability to grow on four out of five examined substrates with moderate growth on BHET. This strain was selected due to the ability to grow fast in examined conditions. According to 16S rRNA sequence analysis, 1204 bp aligned with the highest percentage (99.67%) with *Neobacillus* spp. These species, when present as a part of rhizosphere microbiota, were found to promote plant performance in saline–alkali soil under plastic film mulching. Janakiev et al. isolated and examined species related to this genus from bacteriobiota associated with the gut of *Chironomus riparius* larvae and discovered potential candidates for the formulation of eco-friendly approaches to break down organic pollutants and microplastics within freshwater ecosystems [52]. *Neobacillus* sp. DG 40 showed the ability to grow extremely fast on TSA plates, forming round, smooth, and cream colonies, orange in liquid culture.


**
*Sample 2: Soil sample from plastic buried over 30 years ago*
**


Isolate 69 was selected for further research due to its ability to grow on four out of five offered substrates (all but TPA), especially on Impranil. This strain developed silky-smooth colonies typical for *Bacillus* spp., and BLAST analysis of a 1178 bp sequence suggested the highest similarity with *Priestia megaterium* sp. (basonym: *Bacillus megaterium*) [53]. In the literature, this strain is described as important in biotechnology due to its role as an expression platform for recombinant proteins and vitamins, as well as contributing to bioremediation activities [54]. Meng Tan et al. efficiently used this strain for polystyrene degradation [55]. In addition, this strain is well known as a significant PHB producer under high-salt conditions [56].


**
*Sample 5: Soil contaminated with biopolymers*
**


Isolate 83 exhibited prominent growth on Impranil and PET. After incubation on the TSA plate, it formed round, fuzzy grey colonies. 16S rRNA sequence analysis (1398 bp) indicated 100% similarity to the *Bacillus murallis*. This strain was identified as an LDPE-degrading bacteria and PE waste-colonizing bacteria in the literature [57,58]. 

Isolate 89 exhibited strong growth on PET substrate. Fuzzy-white, round colonies with smooth edges on the TSA plate suggested this isolate belongs to the *Streptomyces* genus, which was further confirmed by 16S rRNA analysis (1378 bp, 100% similarity).

Another strain obtained from this sample, selected for further research, was isolate 90, with the ability to grow on all examined substrates. Opaque, cream-colored and slightly convex colonies, with irregular margins, were observed after incubation on TSA plates. Thus, the 1377 bp 16S sequence showed 100% similarity with *Bacillus circulans* according to BLAST analysis. In a study performed by Saikia et al., this strain was found as part of the community in the gut of the greater wax moth (*Galleria mellonella linnaeus*), able to degrade polycyclic aromatic hydrocarbons, low-density polyethylene, and 2-methylphenanthrene [59]. This strain is also known as a PHA producer [60,61].


**
*Sample 6: Oil-polluted soil*
**


All selected isolates from this sample were identified as *Bacillus* spp. Among the publications employing naturally occurring PET-degrading microorganisms, *Bacillus* spp. was found to be the most frequent bacterial genus [62,63]. All selected strains exhibited the ability to grow on all examined substrates apart from the 114 strain that did not utilize TPA. Isolates 100, 103, and 108 looked similar when grown in TSA and TSB media, but due to their distinctive behavior on MSM plates and in the presence of various substrates, it was presumed that these strains are morphologically similar but not identical. These isolates’ colonies were observed as white, flat, opaque, rough, and wrinkled, with irregular edges on the TSA medium. According to 16S rRNA analysis, these three strains are over 99% similar to each other. The sequence of isolates 100 and 103 (1418 and 1414 bp, respectively) showed 99.86% and 100% similarity to *Bacillus velezensis*, respectively. It is well established that this *Bacillus* strain has the capacity to generate diverse enzymes, including protease, cellulase, amylase, and glucanase. Additionally, it secretes antibacterial substances, effectively inhibiting the growth of pathogenic microorganisms [64,65]. Gui et al. discovered a deep-sea *B. velezensis* with the ability to break down waterborne polyurethane [66]. Also, it has been reported that in the community with *Stenotrophononas maltophilia* and *Acinetobacter radioresistens,* this *Bacillus* strain can affect the decomposition of polystyrene microplastics [66]. Isolate 108 (1333 bp) had the highest similarity to *Bacillus subtilis* (100%). *B. subtilis* is highly resistant to challenging environments and has been used for the secretion of proteins, capable of degrading various pollutants [67]. This quality positions it as a promising microbial tool for biodegradation. Other than PET’s degrading potential [68], Yao et al. reported the ability of *B. subtilis* ATCC6051 to degrade PE [69], and it has been listed as a potential PU degrader [70].

Isolate 109 formed cream-colored colonies, irregular in shape, with undulate margins on the TSA medium. It was established that this strain sequence (1369 bp) has 100% similarity to *Bacillus licheniformis*, a well-established bio-surfactant producer [71]. It was found that in the presence of organoclays, this strain can accelerate the degradation of PLA [72], but also when combined with other *Bacillus* spp., it utilizes PP and is able to degrade PP-PLA blends [73].

When grown on the TSA medium, isolate 111 formed milky-white colonies with undulate edges and a wrinkled surface. Further, 100% similarity to *Bacillus subtilis*, subsp. *stercoris,* and *Bacillus safensis* was detected by BLAST analysis of 16S rRNA sequence (1387 bp). *B. safensis* is known for its ability to degrade PLA [74] but, moreover, it was found by Waquas et al. that this strain can degrade 18.6% LDPE in 30 days [75].

Isolate 114 showed remarkable ability to grow on Impranil-related substrates, exhibiting a halo zone around the irregularly shaped, white colonies with smooth edges. The obtained sequence of this isolate with 1405 bp aligned with *B. licheniformis* and several partial sequences of *Bacillus* spp. isolates (100%).


**
*Sample 7: Landfill in Priboj*
**


All strains selected from sample 7 had the ability to efficiently grow on all examined substrates. Isolates 121 and 142 formed cream, round colonies with smooth edges on TSA medium. Despite being morphologically similar, these strains exhibited different behavior under the examined conditions: 121 expressed profound growth on TPA with a distinctive halo around colonies. Isolate 144, on the other hand, formed white, textured colonies with filiform margins, while 152 had wrinkled grey colonies with smooth margins when grown on TSA. All four of these isolates showed the highest similarity to the *Enterococcus hirae* (100%). *Enterococcus* spp. is rarely connected with plastic degradation research. The ability of *E. faecailis* to degrade polyamide 6 was examined, but the results were not as significant as with *Alcaligene faecalis* [76]. Their ability to colonize LDPE and biopolymers was also investigated with the aim to highlight potential concerns for human health due to the fast pathogens’ biofilm development, not in light of the biodegradation assessment [77]. They have been described as biopolymer producers in the literature [78]. For example, Bhuwal et al. described them as a good candidate for PHB industrial production from cardboard industry wastewater [79]. In this study, isolated *Enterococcus* spp. exhibited an amazing ability to utilize plastic-related substrates as a sole carbon and energy source, making these strains interesting candidates for further research.

Isolate 129 showed exquisite performance during the screen. It was able to grow on all examined substrates, forming a halo zone with the TPA substrate as a sole carbon source. Colonies were disc-shaped with ridges radiating from the center when grown on TSA medium. 16S rRNA sequence (1392 bp) had 100% similarity to *Pseudomonas stutzeri*. *Pseudomonas* spp. is well described in the literature regarding plastic degradation. Highly effective degradation of high-molecular-weight polyethylene glycols (PEGs) achieved through pure cultures of *P. stutzeri* was reported by Obradors et al. in 1991 [80]. This strain was found to degrade the PET monomers terephthalic acid and ethylene glycol [81], but also affect the deterioration of HDPE microplastics [82]. Uefuji et al. reported that this strain secretes PHB depolymerase [83]. 

Isolate 134 was selected due to its excellent performance in the initial screen. Morphologically, this strain behaved as previously described for *Streptomyces* spp.: white, round colonies with smooth edges and textured surface. This was further confirmed by BLAST analysis, where the 16S rRNA sequence (1374 bp) had 99.93% similarity to *Streptomyces* spp.

### 3.3. Phylogenetic Analysis 

The number of plastic-degrading microorganisms reported is rapidly increasing, making it possible to explore the conservation and distribution of presumed plastic-degrading traits across the diverse microbial tree of life. Presumed degraders of conventional high-molecular-weight polymers, like polyamide, polystyrene, polyvinyl chloride, and polypropylene, are distributed extensively across bacterial and fungal branches of the tree of life [84]. As of April 2020, the cumulative count of species documented to possess plastic-degrading capabilities, using specified search terms by Gambarini et al., reached 436 [2], with the initial mention dating back to a publication in 1974 [85]. The same authors reported a total of 16,170 potential plastic degradation orthologs identified across 6000 distinct microbial strains. These strains span 12 different phyla, with 5 of them lacking reported species capable of plastic degradation. In this study, 10 *Bacillus* and *Bacillus*-related strains, 4 *Enterococcus* spp., 4 *Streptomyces* spp., and 1 *Pseudomonas* sp. were isolated from various sources and selected for further research. A phylogenetic tree was constructed, including several bacterial strains that are widely known as plastic degraders in the literature to assess the genetic relationship of isolated stains: *B. subtilis* MZA-75 (polyurethane degrader) [86], *P. aeruginosa* MZA-85 (polyurethane degrader) [87], *Ideonella sakaiensis* (PET degrader), and *Streptomyces* sp. APL3 (polyester-based plastics degrader) [88]. The phylogenetic tree constructed based on neighbour-joining analysis of 16S rRNA clustered 24 examined isolates in three clusters and eight subclusters (Figure 3).

Cluster 1 forms an assemblage of *Bacillus-* and *Enterococcus*-related species, with four distinctive subclusters. In subcluster 1, isolates 100, 103, 108, 109, 111, and 114 closely align with the previously characterized PU degrader, *B. subtilis* MZ-75. This result is in accordance with previously presented screening results, where these isolates exhibited similar behaviour, with an emphasis on PU-related substrate consumption under the examined conditions. Subcluster 2 encompasses isolates 90, 83, 40, and 69. Meanwhile, subcluster 3 consolidates all *Enterococcus* isolates, namely 121, 142, 144, and 152. Notably, isolate 29 takes a divergent path, forming a separate branch as subcluster 4. Cluster 2 gathers *Streptomyces* spp., and two subclusters can be observed: isolate 89 that exhibited excellent performance on PET substrate was closely grouped with *Streptomyces* sp. ALP3 (polyester degrader), while other *Streptomyces* isolates (19, 25, 134) that were more effective on PET monomers were grouped in a separate subcluster. The last cluster contains *Pseudomonas* spp., grouped together with the already well-known PET degrader *I. sakaiensis*, including our isolate 129, capable of using all PET-related substrates, especially TPA, where a halo zone was observed around colonies according to the screening experiment. 

Our study reveals that 20 isolates consisted of four different genera, indicating great potential for species diversity among bacteria exhibiting the ability to degrade polymers. Notably, distinctive patterns emerge concerning the substrate range. While it is essential to approach the interpretation with caution due to site-specific limitations and sample size constraints, these findings are of significant importance in understanding the organisms responsible for the in situ degradation of plastics.

### 3.4. Upcycling Capabilities of Isolated Strains

The primary aim of the BioICEP project is to showcase a sustainable pathway towards a circular economy for plastics. This involves the development of an advanced, energy-efficient, low-carbon, and cost-effective process for transforming waste plastics into high-demand bioproducts and bioplastics (Figure 4). In order to estimate if the selected isolates can be used for plastic waste conversion into high-value products, their emulsification activities and PHA production abilities were assessed. Moreover, pre-treated post-consumer PET (mixture of PET-related substrates) was used to assess PHA production. 

#### 3.4.1. Emulsification Assay

Biosurfactants are gaining prominence as potential natural surfactants, presenting numerous advantages compared to chemical surfactants. These benefits include low toxicity, inherent biodegradability, and ecological acceptability [89]. In our study, the emulsification activity of selected isolates was determined (Figure 5). In comparison to the positive control, the highest (53.57%) emulsification index was for *Aneurinibacillus* sp. DG29. These strains are already reported in the literature as promising biosurfactant producers. For example, *A. aneurinilyticus* was previously described as a potent extracellular and cell-bound lipopeptide producer by Lopez-Prieto et al. [90], while *A. thermoaerophilus* HAK01 isolated from municipal landfill sites was able to produce about 4.9 g/L of lipopeptide at 45 °C [91]. In this study, a notable emulsification index, over 40%, was achieved for *Streptomyces* spp., while the lowest values (below 30%) were obtained from *Neobacillus* sp. DG40, *Bacillus murallis* sp. DG83, and *Bacillus* sp. DG111. Streptomyces spp. has been reported as a successful biosurfactant producer in the literature. For example, *Streptomyces enissocaesilis* HRB1 isolated from the plastic-polluted site emerged as a promising source for the economical production of glycolipid biosurfactants, showcasing potential applications in both biomedical and environmental domains [92]. Zambry et al. reported *Streptomyces* sp. PBD-410L strain with an EI of 46.6% and the ability to effectively produce lipopeptide biosurfactant using palm oil as a sole carbon source [93]. The significance of developing green biosurfactant production methods cannot be overstated, especially considering the critical role of green strategies in the food and health sectors [94]. Utilizing food and plastic wastes can contribute to the generally safe production of biosurfactants; therefore, future studies will be focused on optimization processes, including carbon source exchange with plastic waste. 

#### 3.4.2. PHA Production 

PHAs represent a promising alternative to certain petrochemical-based plastics. These polyesters are synthesized and stored within the cytoplasm of various bacteria and archaea as water-insoluble inclusions. Typically, the production of PHAs occurs when these microbes are cultured under conditions of nutrient limitation, specifically with low concentrations of nitrogen, phosphorus, sulphur, or oxygen, and ample carbon sources. Industries have strategically optimized fermentation conditions to minimize the cost of commercially producing PHAs. In industrial settings, these biodegradable polyesters are obtained through microbial fermentation processes, employing a variety of carbon sources [95,96]. Some examples include the use of discarded plant oils [96], molasses derived from the sugar industry [97], lignocellulosic materials [98], oil palm shells [99], pressed fruit fibre [100], biodiesel waste [101], and waste animal oils [102]. In this study, the ability of selected isolates to produce PHA using pre-treated post-consumer PET as a sole carbon source was examined. Fluorescence was detected in six isolates (Appendix A), and PHA presence and composition were determined by GC analysis (Table 3). 

According to GC analysis, the highest biomass yield was obtained by *Streptomyces* sp. DG19 (1.39 g/L), followed by *Priestia* sp. DG69 (1.06 g/L). *Priestia* sp. DG69 and *Neobacillus* sp. DG40 were the most successful producers with 4.14 and 3.34% of PHA, respectively. The ability of these strains to produce PHAs has been previously recognized. For example, the ability of *Priestia* spp. to produce these molecules using sugarcane molasses as a sole carbon source has been reported, even without a sterilization process prior to inoculation [103,104]. In the study of Sivakuvamar et al., *Streptomyces* sp., isolated from the Bay of Bengal, proved to be a potential source of PHB, utilizing natural carbon sources, such as paddy straw, wheat bran, and rice bran [105]. Co-cultivation of these strains was also reported as successful in PHA production using plant biomass as a carbon source [106].

The composition of produced PHAs depends on several factors, including the type of microorganism, nature of carbon source, and culture conditions. In reality, numerous naturally occurring microorganisms have the capacity to produce PHAs with a diverse monomeric composition, influencing the material properties of the resulting PHA [107]. For this study, additional analysis revealed that a high percentage of 3-hydroxybutyrate (HB) monomers was detected for *Neobacillus* sp. DG40 and *Priestia* sp. DG69 strains: 97 and 25%, respectively (Figure 6). In addition to these, 3-hydroxyvalerate (HV) and 3-hydroxydecanoate (HD) monomers were also detected for these strains in lower percentages. A similar composition was reported by Kenny et al. for the examined *Pseudomonas* spp. [108], with a higher percentage of 3-hydroxypentadecanoate (HDd) in comparison to the results derived in this study. Strain *Streptomyces* sp. DG19 produced 100% of HV monomer, while high percentages of HD monomers were found after cultivation of *Enterococcus hirae* sp. DG142 and *Enterococcus hirae* sp. DG144: 77 and 87%, respectively. High percentages of these monomers were also observed during the cultivation of *Ralstonia eutropha* H16, 80% for HD and 90% of HV monomer, depending on the carbon source type [109].

TPA consumption was estimated via HPLC analysis, and the results are presented in Figure 7. Eight isolates had the ability to use TPA as a sole carbon source in liquid culture; nevertheless, only five produced PHAs. The most effective strains for TPA utilization were *Enterococcus* strains. *Enterococcus hirae* spp. DG142 and DG144 consumed almost 90% of the offered TPA. Interestingly, *Enterococcus hirae* sp. DG121, despite being the only strain capable of forming clearing halos on TPA plates, consumed the same amount of TPA in the time frame examined. Although all of the isolated *Enterococcus* strains possess the necessary enzymatic machinery to utilize TPA, only strain DG121 can secrete these enzymes. Additionally, *Streptomyces* strains DG89 and DG90 consumed >60% of TPA, probably owing to the rich oxidoreduction enzyme arsenal of *Streptomyces* strains [110]. 

Recently there has been a surge in reports highlighting microbial strains with the capability to degrade plastic materials and transform them into valuable products, such as PHA. This development opens possibilities for advancing microbial and enzymatic technologies for the treatment of plastic waste, thereby contributing to the progress of plastics circularity. However, certain challenges persist, requiring additional research. A crucial aspect involves gaining a more profound comprehension of the various pathways of plastic degradation to enhance the development of advanced biotechnological techniques.

## 4. Conclusions

Plastic pollution poses a significant threat to ecosystems and human health, necessitating innovative solutions for its remediation. The findings presented in this research are crucial in advancing our understanding of how microorganisms can be harnessed to combat plastic pollution. By isolating and characterizing strains capable of degrading various plastic substrates, this study lays the foundation for the development of eco-friendly and sustainable strategies for plastic waste management. Several promising strains capable of degrading plastics, particularly PET and PU, were isolated (Table 4). These strains belong to the *Bacillus*, *Enterococcus*, *Streptomyces*, and *Pseudomonas* genera. Phylogenetic analysis demonstrated the diversity of these strains across the microbial tree of life. Additionally, emulsification activity assessment indicated their biosurfactant-producing capabilities making them potentially valuable for various applications, including bioremediation and the food and health industries. Finally, some of the strains exhibited significant potential for PHA production from pre-treated post-consumer PET, contributing to the circular economy for plastics.

This research provides valuable insights into the microbial degradation of plastics, offering a sustainable pathway for plastic waste management. In summary, this research has broad implications for the fields of environmental science, biotechnology, and sustainable waste management, offering a path toward an eco-conscious and resilient future.

## Figures and Tables

**Figure 1 microorganisms-11-02914-f001:**
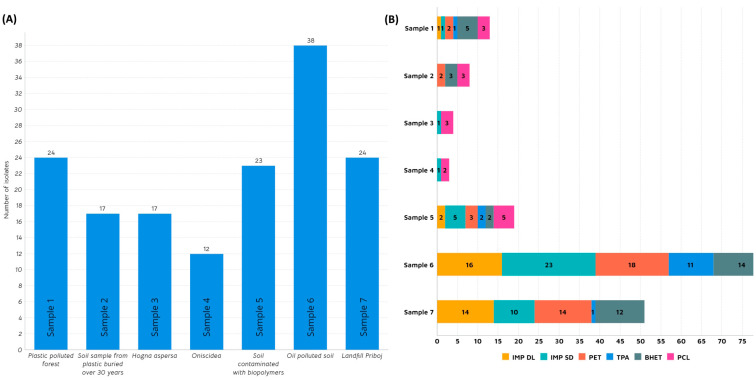
(**A**) Number of isolated strains per sample; (**B**): Number of positive isolates per substrate in agar plate screening: Impranil^®^ DL 2077 (IMP DL), Impranil^®^ DLN-SD (IMP SD), Polyethylene terephthalate (PET), Terephthalic acid (TPA), Bis(2-hydroxyethyl) terephthalate (BHET) and Polycaprolactone (PCL).

**Figure 2 microorganisms-11-02914-f002:**
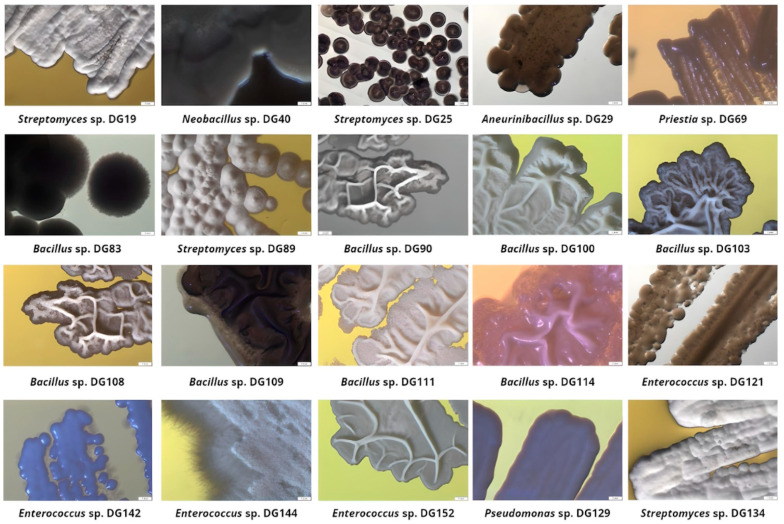
Selected isolates’ colony morphology after incubation on Tryptone Soy Agar plates (magnification 10×).

**Figure 3 microorganisms-11-02914-f003:**
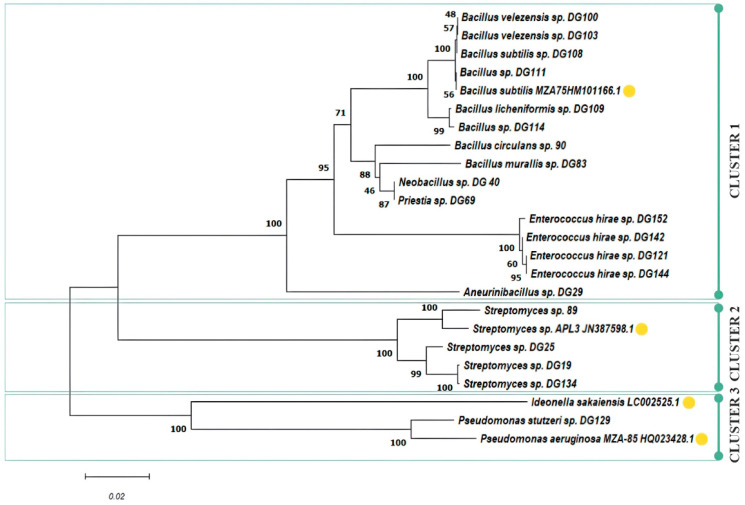
Phylogenetic tree of selected isolates including literature examples of known plastic degraders (yellow circle).

**Figure 4 microorganisms-11-02914-f004:**
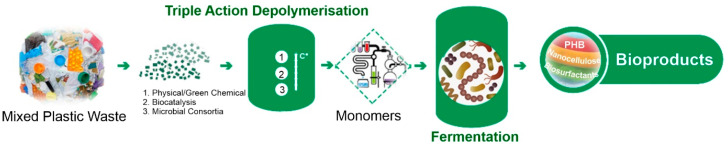
BioICEP project overview.

**Figure 5 microorganisms-11-02914-f005:**
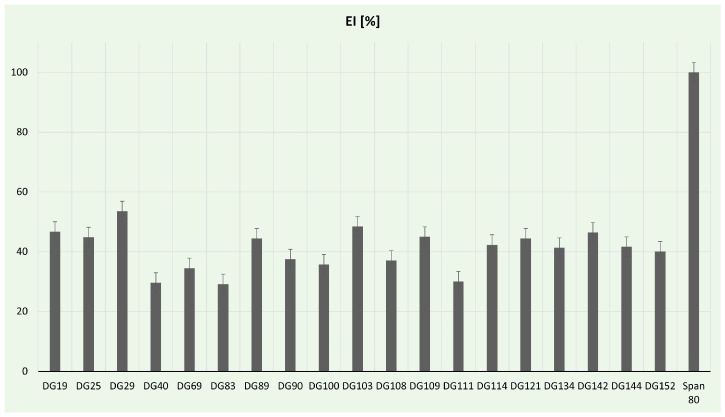
Emulsification assay percentages for tested bacterial strains.

**Figure 6 microorganisms-11-02914-f006:**
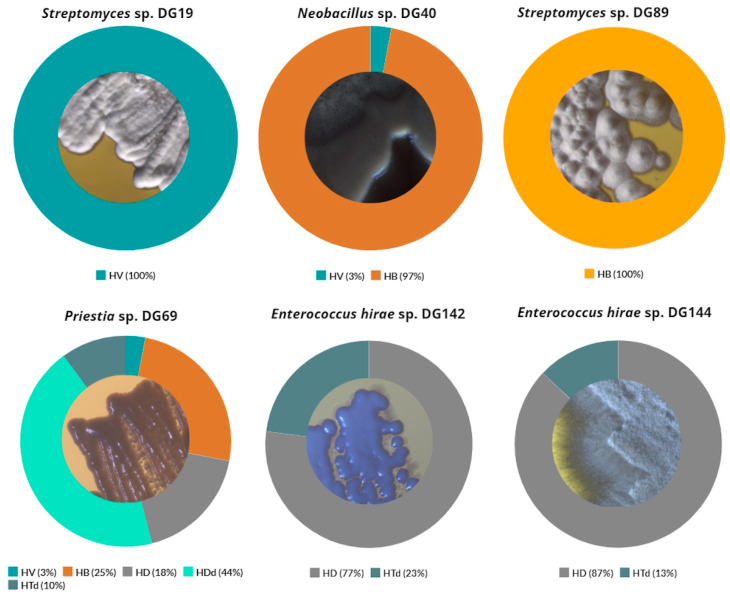
Composition of specific PHA monomers detected after selected isolates cultivation. HB: polyhydroxybutyrate; HV: 3-hydroxyvalerate; HD: 3-hydroxydecanoate; HDd: 3-hydroxypentadecanoate; HTd: 3-hydroxyhexadecanoate.

**Figure 7 microorganisms-11-02914-f007:**
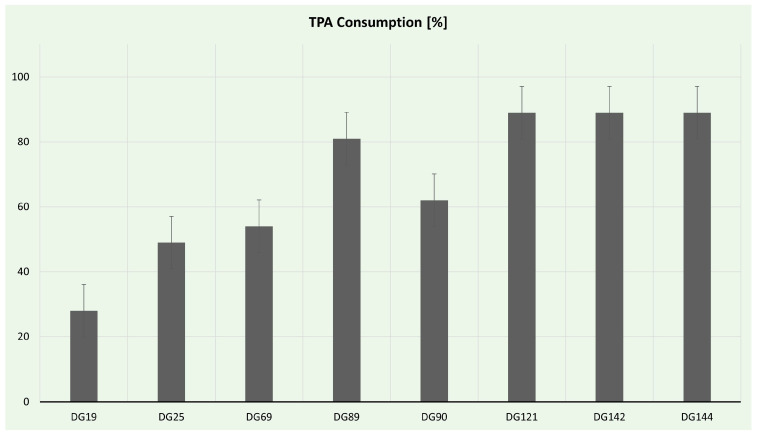
Terephthalic acid consumption during selected isolate cultivation.

**Table 1 microorganisms-11-02914-t001:** Summary of sites used for sample collection for this study.

Sample ID	Collection Date	Description/Criteria	Location	Coordinates	Average Temperature [°C]
S1	20 September 2020	Forest polluted with plastic waste	Athlone, Ireland	53.41991° N 7.88762° W	9–17
S2	1 September 2020	Soil sample with plastic buried 30 years ago	Athlone, Ireland	53.41991° N 8.62351° W	9–17
S3	20 October 2020	*Hogna aspersa*	Athlone, Ireland	53.4239° N 7.9407° W	7–14
S4	20 October 2020	*Oniscidea*	Athlone, Ireland	53.4239° N 7.9407° W	7–14
S5	28 October 2020	Soil contaminated with biopolymers	Athlone, Ireland	53.41991° N 7.88762° W	7–14
S6	10 October 2021	Landfill	Priboj, Serbia	43.5668° N 19.5330° E	5–20
S7	20 January 2021	Oil polluted soil	Poza Rica, Mexico	20.5271° N 97.4629° W	17–25

**Table 2 microorganisms-11-02914-t002:** Selected strains’ performance on examined substrates.

Origin	Isolate	IMP-DL	IMP-SD	PET	TPA	BHET
S1	19					
S2	25					
S2	29					
S1	40					
S4	69					
S5	83					
S5	89					
S5	90					
S7	100					
S7	103					
S7	108					
S7	109					
S7	111					
S7	114					
S6	121					
S6	129					
S6	134					
S6	142					
S6	144					
S6	152					
		* 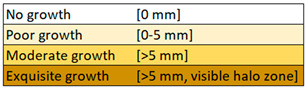 *				

**Table 3 microorganisms-11-02914-t003:** Evaluation of PHA materials obtained after cultivation of selected isolates using residues after PET reactive extrusion as a sole carbon source.

Strain ID	Growth [ΔOD_600nm_]	ΔCDW [g/L]	Fluorescence	PHA [%]	Residual Cell Dry Weight [g/L]
DG19	^(1)^	1.39 ± 0.06	+/−	0.32 ± 0.1	1.337
DG25	^(1)^	0.08 ± 0.1	+/−	n.d.	0.08
DG29	^(1)^	0.62 ± 0.02	−	n.d.	0.62
DG40	0.19	0.44 ± 0.13	+	3.34 ± 0.51	0.425
DG69	0.45	1.06 ± 0.04	+/−	4.14 ± 0.55	1.016
DG83	0.72	0.23 ± 0.11	+	n.d.	0.23
DG89	^(1)^	0.36 ± 0.08	+/−	0.11 ± 0.06	0.359
DG90	0.23	0.13 ± 0.06	+/−	n.d.	0.13
DG97	^(1)^	0.43 ± 0.21	−	n.d.	0.43
DG100	0.38	0	+	n.d.	0
DG103	0.98	0	−	n.d.	0
DG108	0.66	0.12 ± 0.02	−	n.d.	0.12
DG109	0.51	0.11 ± 0.02	−	n.d.	0.11
DG111	0.70	0	+/−	n.d.	0
DG114	0.90	0	+	n.d.	0
DG121	0.42	0.52 ± 0.1	−	n.d.	0.52
DG134	^(1)^	0.9 ± 0.19	−	n.d.	0.9
DG142	0.94	0.27 ± 0.04	+/−	1.26 ± 0.13	0.266
DG144	1.73	0.65 ± 0.1	+/−	2.80 ± 0.56	0.632
DG152	1.35	0.72 ± 0.06	−	n.d.	0.72

^(1)^ Not possible to measure due to cell aggregation; n.d. not detected.

**Table 4 microorganisms-11-02914-t004:** The best-performing isolates’ performance and characteristics.

Isolate	16S rRNA Sequence [bp]	BLAST Analysis Results	Plastic Degradation Ability	Emulsification Index [%]	PHA Production [%]
*Source: Plastic polluted forest*					
DG19	1375	*Streptomyces* sp.	PU, PET, BHET	46.66	0.32 ± 0.1
DG40	1204	*Neobacillus* sp.	PET, BHET	29.62	3.34 ± 0.51
*Source: Soil contaminated with biopolymers*					
DG89	1378	*Streptomyces* sp.	PU, PET	44.44	0.11 ± 0.06
*Source: Soil sample from plastic buried over 30 years*					
DG69	1178	*Priestia* sp.	PU, PET	34.48	4.14 ± 0.55
*Source: Landfill Priboj*					
DG142	1296	*Enterococcus* sp.	PU, PET, TPA	46.42	1.26 ± 0.13
DG144	1367	*Enterococcus* sp.	PU, PET, TPA	41.66	2.80 ± 0.56

## Data Availability

Data are contained within the article and Appendix A.

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
