# Peer review of "Exploring Microorganisms from Plastic-Polluted Sites: Unveiling Plastic Degradation and PHA Production Potential"

_microorganisms, 2023, doi:10.3390/microorganisms11122914_

Round 1

Reviewer 1 Report

Comments and Suggestions for Authors

This manuscript focuses on microorganisms from plastic-polluted sites, revealing their ability to degrade plastics and produce PHA potential. The general evaluation of the manuscript is as follows: (1) The research on plastics and PHA-related enzymes needs to be strengthened. (2) The results and discussion section has a lot of descriptions based on observations and lacks in-depth discussion and insights. (3) The mechanism of plastic degradation by isolated strains was not revealed.

Abstract:

Line 21: The changes in microbial communities and their diversity were not discussed and analyzed in the results, only the isolated strains were studied.

Line 19-35: There is a lack of refining of experimental highlights. The practical implication of the results remains unclear. The current version of the abstract did not well present the content of this work.

Introduction:

Line 38-122: The introduction is redundant, does not closely integrate with the article's topic, the logic is not smooth, and needs to be condensed.

Line 42-60: The introduction mentioned the degrading enzymes of plastics, and the results did not identify the degrading enzymes.

Line 62-74: PHA-related enzymes were not analyzed in the results.

Materials and Methods:

Line 125-128: What was the reason for choosing these sampling sites, such as plastic-polluted sites, landfills, oil-polluted sites, but also arthropods (isopods and arachnids)?

Line 127: What is the sampling of arthropods (isopoda and arachnids) meant to show? How are arthropod samples sequenced?

Line 132-143: Change “g/l” to “g/L”, and check the unit format of the full manuscript.

Line 132-143: Why choose these plastic monomers as the sole carbon source, such as terephthalic acid (TPA), bis(2-hy-droxyethyl) terephthalate (BHET), polyethylene terephthalate (PET), anionic aliphatic polyester-polyurethane dispersion and anionic polycarbonate polyurethane dispersion? What form of plastic is used as a carbon source? Micro or nanoscale?

Line 151: Change “2.4.16. S rDNA sequencing” to “2.4. 16S rDNA sequencing”.

Line 165: Unify “16S rDNA” and “16S rRNA” in manuscripts.

Results and Discussion: Authors need to provide a deeper understanding of data and specific discussions with literature.

Table 2: “TPA” is miswritten as “TA”.

Line 310-319: The selection and identification of isolated strains are described in detail. The paper should emphasize the mechanism of the plastic degradation process, and it is suggested to rearrange the structure of the paper.

Line 360-361: What accounts for significant phylogenetic diversity? What is the proof of significance?

Line 367: How to evaluate the degradation performance of isolated strains on plastics? The by-products after plastic degradation should be determined and the degradation pathways of different plastics should be revealed.

Line 377-388: This paragraph sounds like a review, not an experimental article.

Line 282, 411: Error bars of parallel experimental results were added to Figure 1 and Figure 4. All experimental results of this study were statistically analyzed.

Conclusions:

Line 469-487: It is suggested to summarize the experimental conclusions.

References:

The format is confusing and some information is missing.

Reviewer 2 Report

Comments and Suggestions for Authors

The manuscript by Herrera and co-authors explores the potential of microorganisms from plastic-polluted sites to convert plastics into value-added bioproducts.

Major comments

1. The introduction section is too long. This reviewer suggests it should be shortened.

2. Please style all scientific terminology in italics - please correct the text and the reference list.

Minor comments

L 67-70. PhaZ (depolymerases) can be cited in this portion of the text with adequate references. Please rewrite.

L 93-95. Please cite which bioproducts are the target molecules in this project.

L 96-100. This reviewer considers important to mention that not all the bio based plastics are biodegradable; and that the PHA are both bio based and biodegradable. Please rewrite.

L 124. Please add more information on the choice of sampling sites to the introduction section.

L 127. Were arthropods selected as a sampling site? Please rewrite for clarity.

L 144. Were all the carbon sources tested at 3 or 10 g/L? Please rewrite indicating the used concentration(s) for each tested carbon source. 

L 145-149. Was any other carbon source used a positive control? Please clarify.

L 151. 16S rDNA? please correct.

L 209. LB does not stand for Luria Bertani - https://wiki.rice.edu/confluence/pages/viewpage.action?pageId=33294746 Please correct.

L 225-226. Please note that relying on OD600 measurements for constructing growth curves is unreliable when dealing with PHA-accumulating strains. - https://doi.org/10.1139/cjm-2019-0342

L 264-265. Kindly provide some information regarding the metabolic regulation necessary for utilizing these molecules as the primary carbon source. Consider that other classic carbon sources (e.g. carbohydrates) are present in this environments.

L 283. Fig 1A would be improved if the name of the sampling site is placed on the X axis.

L 317. Figure 2 - is TA = TPA? Please correct.

L 318. Were the Figures presented from L318 obtained in this study? Please number these figures and reference all of them in the text.

L 319. Please add a table summarizing the 16s results.

L 341. Please use a high quality image.

L 410. Please include error bars to Fig. 4.

L 438. Please add the residual cell dry weight (CDW minus PHA in g/L) to Table 3.

L 439. Caption is missing for (1), n.d., +/-. Please correct.

L 439. Were growth rates calculated from the measured OD600? If yes, please consider including this information in Table 3.

L 440. Also, please provide the growth curves in the supplementary material.

L 442. PHV or the 3HV monomer? Same for PHD (3HD?). Please clarify and correct the text.

L 444. Please include literature information regarding the monomeric composition of the PHA produced in this study. Has this composition been previously reported for these strains? This information would enrich the discussion section.

L 448. Please define HB, HV, HD, HDd, HTd in the main text and in the Figure caption.

L 460. Please add error bars to Fig. 6.

L 461-468. Please summarize the most promising strain(s), describing which substrates they could use and which monomers could be incorporated into the final PHA.

Comments on the Quality of English Language

Minor edits are required.

Reviewer 3 Report

Comments and Suggestions for Authors

The manuscript by Herrera et al. described the isolation of several bacterial strains from plastic contaminated sites. Some of the isolates were found able to degrade/convert plastic-related substrates and produce PHA. The study has potential values. However, the current result/evidence is insufficient in many ways, and the data presentation needs to be improved. Please consider the comments/suggestions below for improvement.

1.     The authors need to show that your findings/isolates can indeed be potentially used for the bioprocesses shown in Figure 3. That means 1) can any of your isolates degrade or convert real/commercial plastic beside the plastic-related substrates you used? 2) Can any of your isolates do PET depolymerization for PHA production. As the substrate you used for PHA production was chemically produced via RES.

2.     What is the novelty and significance of your isolates compared to other known plastic-degraders? Did you find novel species of plastic-degraders? What distinct features do your isolates have? These need to be highlighted in the manuscript.

3.     The image of agar-grown cells of the isolates is not sufficient to show their morphology. TEM/SEM images are needed.      

4.     The result in Table 2 is very un-scientific! What do you mean by poor, moderate, exquisite growth? What are the growth rate and degradation rate for each strain?

5.     Results in Figures 4-6 and Table 3 lack replicates/error bars.

6.     Make a table list all the isolates and include the information such as taxonomic identification, plastic-related substrates used, emulsification potential, PHA production… Such table can provide a clear guidance for the readers of this paper.

Comments on the Quality of English Language

English phrasing/expression need to be improved. Also check the grammar and typos.

Round 2

Reviewer 1 Report

Comments and Suggestions for Authors

The authors responded to the comments and the manuscript met the requirements for acceptable publication.

Author Response

We would like to thank you for all your comments in Round 1. Review process significantly improved our manuscript and we are very happy to hear you are satisfied with the new version.

Kind regards!

Reviewer 3 Report

Comments and Suggestions for Authors

The manuscript has been improved and the comments were addressed fine. I don't have more comments.

Please correct the followings:

Be consistent with "16S rRNA gene" in the text. As for now sometimes it's "16S rRNA"  sometimes "16S rRNA gene"

line457 (Figure 3). Phylogenetic tree, not "Phylogenetic three"

Author Response

First of all, we would like to thank you for all your comments in the first round. Review process really significantly improved our manuscript. Also, thank you for noticing details in the second review round, it really shows your dedication and thoughtful reading. We corrected our manuscript as suggested in your comments. 

Kind regards!